# Heteroaromatic Diazirines Are Essential Building Blocks for Material and Medicinal Chemistry

**DOI:** 10.3390/molecules28031408

**Published:** 2023-02-01

**Authors:** Yuta Murai, Makoto Hashimoto

**Affiliations:** 1Graduate School of Life Science, Hokkaido University, Kita 21, Nishi 11, Kita-ku, Sapporo 001-0021, Japan; 2Faculty of Advanced Life Science, Hokkaido University, Kita 21, Nishi 11, Kita-ku, Sapporo 001-0021, Japan; 3Division of Applied Bioscience, Graduate School of Agriculture, Hokkaido University, Kita 9, Nishi 9, Kita-ku, Sapporo 060-8589, Japan

**Keywords:** diazirine, heteroaromatics, medicinal chemistry, photoaffinity labeling

## Abstract

In materials (polymer) science and medicinal chemistry, heteroaromatic derivatives play the role of the central skeleton in development of novel devices and discovery of new drugs. On the other hand, (3-trifluoromethyl)phenyldiazirine (TPD) is a crucial chemical method for understanding biological processes such as ligand–receptor, nucleic acid–protein, lipid–protein, and protein–protein interactions. In particular, use of TPD has increased in recent materials science to create novel electric and polymer devices with comparative ease and reduced costs. Therefore, a combination of heteroaromatics and (3-trifluoromethyl)diazirine is a promising option for creating better materials and elucidating the unknown mechanisms of action of bioactive heteroaromatic compounds. In this review, a comprehensive synthesis of (3-trifluoromethyl)diazirine-substituted heteroaromatics is described.

## 1. Introduction

Heteroaromatic compounds are essential building blocks in a wide variety of functional molecules, including organic materials [1], electric devices [2], pharmaceuticals [3], and natural products [4]. For example, installation of crosslinks in (heteroaromatic) polymers can improve mechanical strength and corrosion resistance. However, it is difficult to install crosslinks into ingredients of a functional polymer in a well-controlled manner due to the lack of functional groups required for coupling. In recent materials science, photocrosslinking has been one of the essential chemical techniques used to make polymer networks into functional materials in order to increase thermal stability, shock resistance, corrosion resistance, and tensile strength without the particular functional groups required for coupling [5,6]. On the other hand, photoaffinity labeling (PAL) has also been demonstrated to identify ligand-binding biomolecules in a variety of drug-discovery fields. Its site mapping and protein–protein interaction are visualized through live-cell imaging of photoinduced crosslinkage [7,8,9,10]. Especially in PAL experiments, selection of photophores is an important factor for the performance of an efficient photolabeling reaction. Three common photophores, arylazide [11,12], benzophenone [13], and trifluoromethyldiazirine [14], have been used and contributed to PAL. To the best of our knowledge, trifluoromethyldiazirine is a useful photophore because it has advantages such as its relatively small size, stability, and low rate of rearrangement and generates a highly reactive carbene with a longer wavelength (≈365 nm), which reduces damage to polymer materials and biomolecules. Therefore, a combination of heteroaromatics and trifluoromethyldiazirine is a rational strategy for preparation of functional polymer materials and for revealing the unexplained mechanisms and target biomolecules of bioactive heteroaromatic compounds. In this review, a comprehensive synthesis of (3-trifluoromethyl)diazirine-substituted heteroaromatics and their applications are described.

## 2. Diazirinyl-Substituted Pyridines and Pyrimidines

Recently, use of (3-trifluoromethyl)phenyldiazirine (TPD) has been widely increasing in materials science for creation of crosslinking of component parts of organic light-emitting diodes [15,16], for primers for fiber-reinforced polymer composites [17], and for patterning of wearable elastic circuits [18]. Despite the success of TPD as a photocross-coupling agent, in order to establish functional materials more efficiently and easily with versatile photocrosslinking, there is a need to improve the photolabeling performance of TPD. This includes factors such as stability against ambient light and water solubility to enable reactions in aqueous solutions. Kumar et al. designed and synthesized novel 3-pyridyl and 3-pyrimidyl-substituted 3-trifluoromethyl-diazirines 11 and 12, which possess stability under ambient light and have aqueous solubility [19]. As shown in Figure 1, 5-bromo pyridyl compound 1 and pyrimidyl compound 2 were subjected to the protection of primary alcohol with silyl groups (*tert*-butyldimethylsilyl, TBS; *tert*-butyldiphenylsilyl, TBDPS); subsequently, trifluoroacetylation of compounds 3 and 4 was conducted with *n*-BuLi and methyl trifluoroacetate to obtain 5 and 6 with a moderate yield. Next, the trifluoroacetyl groups were converted to oximes, followed by tosylation in an appropriate manner for each compound. Tosyl oximes 7 and 8 underwent the addition of ammonia (liquid) to give diaziridines 9 and 10 with a good yield. Finally, the desired products, diazirines 11 and 12, were obtained through oxidation of the diaziridinyl moiety with silver oxide and deprotection of the silyl group with tetrabutylammonium fluoride (TBAF).

The photoactivation and the ambient light stability of synthesized diazirinyl compounds 11 and 12 were tested and compared to those of (4-(3-(trifluoromethyl)-3*H*-diazirin-3-yl)phenyl)methanol 13. The photoreactive kinetics of these three compounds were calculated through photoactivation of a solution of the photolabel in methanol-*d*_4_ with UV light and measurement of the compound change using ^19^F NMR over time. Neither electron-withdrawing pyridine nor pyrimidine rings affected the ratio of generation of carbene from a diazirine, and they indicated the same photoreactive efficiency as in compound 13. Moreover, the ambient light stability of 11 and 12 was investigated. Compound 13 had already demonstrated significant photodecomposition after seven days with ambient light exposure. In contrast, compounds 11 and 12 were negligibly photodecomposed. After exposure of the probes to ambient light for a period of one month, only 27% of compound 13 remained, whereas 79% of compound 11 and 90% of compound 12 remained (Table 1).

Green chemistry in materials science is one of the most vital challenges: for example, replacement of organic solvents, success of organic reactions in aqueous solutions, etc. The aqueous-solubility enhancements of compounds 11 and 12 were demonstrated, and it was confirmed, using a HPLC-based assay, that the 11 and 12 derivatives were 100–7500 times more soluble than the 13 derivatives at pH = 7.4 and 5.0 [20] (Table 2).

Therefore, 3-pyridyl and 3-pyrimidyl-substituted 3-trifluoromethyl-diazirines 11 and 12 demonstrated significant ambient-light-stability and aqueous-solubility improvements over the conventional 3-trifluoromethyl-3-aryldiazirines. These physicochemical properties could contribute huge advantages, and not only for photolabeling experiments in materials science.

## 3. Diazirinyl-Substituted Benzimidazoles

Benzimidazole and imidazole derivatives are medicinally used as anticancer [21,22,23,24,25], antioxidant [26,27], anti-inflammatory [28,29], anticoagulant [30,31], anthelmintic, or opioid products, and for analgesic activity [32]. Therefore, in order to discover better medicinal reagents or drugs, it is important to identify the target biomolecules of benzimidazole and imidazole derivatives and understand the mechanism of action of their pharmacological effects. Diazirinyl-substituted benzimidazole was reported by Raimer et al., and its chemical properties, thermal stability, and photoreactivity have been demonstrated [33]. The starting material, 14, was prepared following a previous report [34], then converted to oxime, tosyloxime, diaziridine 15, and diazirine 16 in four steps according to standard protocol (Figure 2). Moreover, the synthesis of diazirinyl-substituted imidazoles was also attempted, but only diaziridines were able to be reached due to the violent decomposition of the diazirinyl imidazoles with self-ignition.

Next, the physical properties of diazirinyl-substituted benzimidazole 16 were investigated. The photoreaction of 16 in EtOH was slow, and ethoxy adduct 17 was formed, with a moderate yield (46%). Interestingly, in the case of decreasing the ethanol ratio to one equivalent of 16 in the presence of CH_2_Cl_2_, the reaction proceeded more quickly and gave a higher yield (60%; Figure 3). This observation was already reported, identifying which solvent effect on the carbene reaction explained its different reactivity [35,36,37,38]. Furthermore, compound 16 was proved to have thermal stability at up to 88 °C in differential-scanning calorimetry measurement. Therefore, diazirinyl-substituted benzimidazole 16 could be the backbone of a reliable PAL probe and contribute to new drug discovery through identification of target molecules of benzimidazole-containing bioactive substances.

## 4. Diazirinyl-Substituted Pyrazoles

Pyrazoles, five-membered heterocyclic compounds including two nitrogen atoms, are an important class of compounds for drug development, with a wide application in medicinal chemistry [39,40,41]. Pyrazole derivatives exhibit diverse targets and resistance: for example, to cancer [42,43], acquired immunodeficiency syndrome [44,45], tubercular [46], insects [47], etc. Furthermore, 5-Amino-1-[2,6-dichloro-4-(trifluoromethyl)phenyl]-3-cyano-4-(trifluoromethyl)sulfinylpyrazole (fipronil, Figure 4), a chemical pesticide, is a widely used broad-spectrum insecticide. Fipronil, as a noncompetitive blocker, inhibits the γ-aminobutyric acid (GABA)-gated chloride-ion channel receptor. Furthermore, within the GABA receptor family, dysfunction of the Type A GABA receptor leads to neurological disorders and mental illnesses in humans; therefore, the GABA receptor is a drug-discovery target for these disorders [48,49,50]. Moreover, understanding fipronil’s noncompetitive binding site in the GABA-gated chloride channel might be a fascinating matter for medicinal chemistry.

For this purpose, fipronil-based photoaffinity probe 18 (Figure 4) was prepared by the Casida J.E. group [51,52]. Its synthesis was started from commercially available pyrazole 19, reacted with iodine and ceric ammonium nitrate to produce 4-iodopyrazole 20. Compound 21 was obtained via the nucleophilic aromatic substitution of 20 with potassium carbonate in DMF at 100 °C with a quantitative yield. Subsequently, an iodine–magnesium exchange of 21 was performed with *i*-propylmagnesium chloride, followed by nucleophilic substitution with *N*-(trifluoroacetyl)piperidine to obtain 4-trifluoroacetylpyrazole 22. Finally, compound 18 was prepared according to a previous synthetic method [53,54,55] (Figure 4).

The use of radioisotopes incorporated in a PAL probe is an efficient method for detection of labeled molecules because of the radioisotopes high sensitivity. Thus, the incorporation of tritium into fipronil-based photoaffinity probe 18 was also attempted. Initially, iodine incorporation into compound 18 or 22 was attempted several times via ortho lithiation; however, the desired products could not be obtained. Conversely, compound 25, which was prepared through the reduction of 22 with NaBH_4_, was smoothly subjected to iodination under standard lithiation conditions [56]. Compound 24 was reoxidized back to trifluoroacetyl 25, with the Dess–Martin reagent [57,58], which was followed by conversion to diazirine 26. Next, reduction of diiodoarene 26 with tritium gas, 10% Pd/C, and triethylamine in ethyl acetate was conducted to obtain tritium-labeled, fipronil-based photoaffinity probe 27 (Figure 5). The binding potency of 18 for the GABA receptor was also evaluated through competitive inhibition with 4′-ethynyl-4-[2,3-^3^H_2_]propylbicycloorthobenzoate, and the molar concentrations for 50% inhibition via 18 indicated approximately 2 nM. Therefore, compounds 18 and 27 could be suitable mimics of fipronil for use in photoaffinity labeling.

## 5. Diazirinyl-Substituted Benzoxazolinone

Benzoxazolinones 28, 29, and 30 (Table 3), isolated from light-grown maize (*Zea mays* L.) shoots, have the potential to interfere with auxin behavior or inhibit the auxin receptor [59]. Kosemura, S. et al. reported that the structure–activity relationships of benzoxazolinones 31 and 32 (Table 3) were related to auxin-induced growth and auxin-binding protein [60]. The precise mechanism through which this bioactivity is triggered via benzoxazolinones has remained unknown. To address this subject, Kosemura, S. et al. tried synthesizing two photolabile benzoxazolinone analogues, 33 and 34, and evaluated their photoreactivities [61].

Photoreactive compound 42 was prepared from starting material 3-bromoanisole 35 according to a previous method [62,63]. Briefly, compound 35 was converted to the Grignard reagent treated with magnesium, followed by nucleophilic substitution with *N*-(trifluoroacetyl)piperidine to obtain 3-trifluoroacetylanisole 36. Compound 40 was prepared from 36, following the general diazirine synthetic method. Subsequently, nitration of 40 with fuming nitric acid and demethylation of 41 with BBr_3_ were conducted to obtain compound 42 with a moderate yield, 42%, in two steps. The reduction of the nitro group in 42 was carried out with Na_2_S_2_O_4_ while diazirine reduction was avoided as much as possible. Finally, Compound 43 was subjected to phenyl carbamation 44, followed by intramolecular cyclization to obtain the desired product, 33, with 14% in three steps (Figure 6).

Moreover, photoreactive compound 34 was synthesized with the following procedure: Phenolic alcohol 45 was protected with chloromethyl methyl ether (MOMCl), followed by bromination with *N*-bromosuccinimide (NBS), to obtain compound 46. Trifluoroacetylation of 46 was carried out through a bromine–lithium exchange with *n*-BuLi, followed by nucleophilic substitution with ethyl trifluoroacetate, to obtain compound 47. Diazirine 48 was prepared according to the same procedure to construct 40 from 36. Nitration of 48 with fuming nitric acid was conducted at −72 °C to avoid the decomposition of the diazirine moiety. Compound 49 was deprotected with acidic hydrolysis 50 and then protected again with acetic anhydride to obtain compound 51. The nitro group of 51 was subjected to reduction with Na_2_S_2_O_4_ within 5 min, followed by carbamation with phenyl chloroformate, then treated with sodium hydroxide to obtain the desired diazirine, 34 (32% in three steps) (Figure 7).

Next, diazirinyl benzoxazolinones 33 and 34 were evaluated as to whether they had suitable characteristics for photoaffinity-labeling reagents. The photoirradiation of compound 33 (1 mM in methanol) was smoothly carried out with a black light (12 W) to produce methanol adduct 53 with a moderate yield. The half-life (*t*_1/2_) of 33 was calculated to be approximately 16 min, whereas the photoreactive kinetics of 34 with the black light (12 W) were much slower than those of 33. Using a 500 W high-pressure mercury lamp for photolysis of 34, compound 54 was produced, and the half-life (*t*_1/2_) of 34 was 6 min (Figure 8). Therefore, both of the compounds have suitable characteristics for photoaffinity-labeling reagents.

## 6. Diazirinyl-Substituted Benzoxazole

Duchenne muscular dystrophy (DMD), caused by loss-of-function mutations in the dystrophin gene, leads to progressive muscle degeneration and results in heart and respiratory failure [64,65]. Although quality of life and longevity have been improved with developments in the clinical standard of care [66,67,68], there is no complete treatment available for DMD. Recently, ezutromid (55) has been developed as a utrophin modulator and demonstrated reduced muscle-fiber damage and increased levels of utrophin after 24 trial weeks [69]. However, due to an administration effect, ezutromid could not retain this effect for a long period, and the mechanism of action of ezutromid is unknown. The development of ezutromid was discontinued. Therefore, to elucidate the mechanism of action of ezutromid for these effects and discover new drugs for DMD, Wilkinson, I.V.L. et al. synthesized photoreactive ezutromid derivatives 56 and 57 [70] for use in affinity-based protein profiling (ABPP) [71,72,73]. As a photophore of 56 and 57, 3-trifluoromethyldiazirine was chosen to replace the ethylsulfonyl group of ezutromid because of the similarity in the electronics of the diazirine and sulfonyl groups. The naphthyl moiety was replaced with alkynyl-phenyl substituents for installation of detection tags with click chemistry. The synthetic scheme of photoreactive compounds 56 and 57 is shown in Figure 9, beginning with microwave-assisted benzoxazole cyclization 60 from 3-amino-4-hydroxybenzoic acid 58, and the corresponding acid chloride, 59, without purification [74]. The carboxylic acid was subjected to conversion of Weinreb amides 61 and 62, followed by trifluoroacetylation with the Ruppert–Prakash reagent to obtain 63 [75]. Diazirine derivatives 64 and 65 were introduced according to the general method [76]. Finally, Sonogashira couplings with TMS-acetylene were followed by TMS deprotection to obtain the desired products, 56 and 57, with a moderate yield (50–70%).

Furthermore, compounds 56 and 57 were applied to ABPP and found to bind the aryl hydrocarbon receptor (AhR). As a result, ezutromid was revealed as a novel AhR antagonist, inhibiting nuclear translocation and downregulating AhR-responsive genes such as AhRR and Cyp1b1. Therefore, this study could pave the way for the first target-based drug discovery in DMD treatment as well as provide a biomarker for future clinical trials.

## 7. Diazirinyl-Substituted (Benzo)thiophene

Thiophene is a five-membered, sulfur-containing heteroaromatic ring commonly used as a building block in the field of medicinal chemistry. In particular, thiophene derivatives are expected to possess a wide range of therapeutic properties, such as antitumor [77], anti-inflammatory [78], antiarrhythmic, antianxiety [79], and antifungal [80] effects and kinase inhibition [81,82]. Thus, evaluation and mechanism elucidation of novel thiophene moieties with wider therapeutic activity are a topic of interest for the medicinal chemist to synthesize and investigate new structural prototypes with more effective pharmacological activity. Oncodazole (methyl [5-(2-thienylcarbonyl)-1*H*-benzimidazol-2-yl]carbamate) 66, composed of a thiophen skeleton, is a tubulin-binding agent that has potent anthelmintic [83] and antifungal [84] activities. In particular, tubulin binding via drugs that cause disruption or hyperstabilization of the mitotic apparatus is presently an area of great interest. Thus, in order to characterize the interaction of 66 with tubulin, Ladd, D.L. et al. evaluated diazirinyl oncodazole (methyl [5-(2-thienylcarbonyl)-1*H*-benzimidazol-2-yl]carbamate) 67 [85]. They placed the diazirine function in the 4′-position on the basis of a systematic study of oncodazole derivatives because the position could be substituted without loss of biological activity.

Firstly, compound 68 was converted to acid chloride 69 with thionyl chloride, followed by a Friedel-Crafts reaction with anisole and without purification to obtain compound 70 with a quantitative yield. After protection of the carbonyl group of 70 with ethylene glycol, nucleophilic substitution was carried out with *N*-(trifluoroacetyl)piperidine through a lithium–bromo exchange on 71 to obtain trifluoroacetyl derivative 72. Subsequently, compound 73 was produced via acid hydrolysis of ethylene ketal; then, a reaction of 73 with nitric and sulfuric acids produced the appropriately substituted nitro derivative 74.

Next, compound 74 was converted into diazirinyl derivative 75 through the general TPD synthetic procedure. The methoxy group of 75 was substituted to an amine group with ammonia (liquid) with a good yield; then, the nitro group of 76 was reduced with sodium hydrosulfite, with diazirine moiety degradation avoided as much as possible. Finally, both diamine groups were subjected to condensation with bis(methoxycarbonyl)-*S*-methylisothiourea catalyzed with *p*-toluenesulfonic acid to obtain the desired compound, 67 (Figure 10). Furthermore, the photoreactivity and affinity of diazirinyl derivative 67 to tubulin were evaluated. The photolysis of compound 67 (0.5 mM in methanol) was carried out with a black light under 45 °C for 20 min. The half-life (*t*_1/2_) of 67 was calculated to be approximately 21 s. In addition, the relative affinity of 67 to tubulin was determined through competitive-equilibrium binding-assay displacement of ^3^H-labeled oncodazole. In the result, IC_50_ values of 5.7 ± 1.9 μM for 66 and 8.3 ± 3.0 μM for 67 were obtained; therefore, compound 67 has potential as a photoaffinity-labeling reagent.

Furthermore, the benzothiophene scaffold is a capable moiety in drug discovery because it exhibits various biological activities such as antimicrobial [86], anticancer [87,88], and anti-inflammatory [89] effects and many more. Therefore, diazirinyl benzothiophene (81) was synthesized, according to a general TPD synthetic method, by Wang, J. and Sheridan, R.S. [90]. Briefly, benzothiophene-2-trifluoromethyl ketone 77 was oximated with hydroxylamine hydrochloride in pyridine 78, followed by tosylation with tosyl chloride in CH_2_Cl_2_ with triethylamine 79. Subsequently, the tosyl oxime moiety was subjected to diaziridinylation with liquid ammonia under −65 °C. Finally, diaziridine 80 was oxidized to the desired diazirine, 81, with iodine, which resulted in a total yield of 26% for proceeding (Figure 11).

## 8. Diazirinyl-Substituted Coumarin

Coumarin (1,2-benzopyrone) and its derivatives are widely available in nature and exhibit various biological activities such as antitumor [91], anti-HIV [92,93,94,95], and anti-inflammatory effects [96,97] as well as resistance to triglyceride accumulation [98] and central-nervous-system stimulant effects [99]. In addition to this, coumarin–metal complexes have also attracted attention due to their interesting fluorescent properties as sensors or probes [100,101]. Therefore, there are broad possible uses of coumarin derivatives in materials and medicinal-chemistry scientific research.

In PAL, generally, there are some problems while identification of the cross-linked target molecules due to involves complexity of reaction mixtures and resulting in small quantity of the photo-labelled molecules. Therefore, photolabeled molecules require enrichment with detection tags such as the biotin–avidin system, fluorous tagging [102,103], or clickable groups [104]. However, these PAL probes are each composed of a photophore, a ligand, and a detection tag via a branching structure, and the affinity of the PAL probe towards target molecules tends to reduce because of bulky and unstable fluorophores. Recently, to overcome these problems, Tomohiro, T. et al. have developed a diazirinyl coumarin as a photocrosslinker with a masked fluorogenic beacon [105]. In this section, synthesis of diazirinyl coumarin derivatives and applications for PAL are presented. The synthetic schemes of diazirinyl coumarin derivatives 90a and 90b are described in Figure 12. Initially, 1-chloro-3,5-dimethoxyphenylbenzene was converted to the Grignard reagent through treatment with magnesium and 1,2-dibromoethane, followed by nucleophilic substitution with 2,2,2-trifluoro-1-(piperidin-1-yl)ethanone to obtain compound 83 with a good yield. Diazirinyl compound 87 was prepared according to the general diazirine synthetic method. Subsequently, compound 87 was subjected to formylation with TiCl_4_ and dichloromethyl methyl ether to obtain two regioisomers, 88a and 88b, which were able to be separated with silica column chromatography. The methoxy group of each compound 88 was removed with BBr_3_; then, compounds 89a and 89b were treated with Meldrum’s acid (2,2-dimethyl-1,3-dioxane-4,6-dione) to obtain the desired diazirinyl coumarin derivatives, 90a and 90b (Figure 12).

Next, derivatives of both compounds 90a and 90b (20 mM in CD_3_OD) were subjected to photoreaction with a 250 W black-light lamp at room temperature for 1 h, monitored, with ^1^H and ^19^F-NMR. The compound-90a derivative was smoothly photolyzed to generate a CD_3_OD adduct without production of a diazo compound, whereas the compound-90b derivative exhibited different phenomena. The rate constant of photolysis of the 90b derivative was calculated as less than one twentieth of the value of the compound-90a derivative. In addition to this, the coumarin fluorescence of the photolyzed 90b derivative was weaker than that of the 90a derivative. On the basis of these data, compound 90a was determined to be suitable as a crosslinker unit for further fluorogenic labeling of proteins.

Furthermore, the versatility of compound 90a as a photoaffinity probe with geldanamycin (GA), which is a potent inhibitor of heat-shock protein 90 (Hsp90), was demonstrated. A photoactivatable GA probe 91 (Figure 1) was prepared according to the synthetic method in a previous report [106] and demonstrated an inhibitory effect on ATPase activity through competition for ATP binding (dissociation constant (*K*_d_) = 1.2 μM) [107]. As a result of Hsp90 PAL with probe 91, photolabeled Hsp90 was detected through measurement of coumarin fluorescence, which increased in an irradiation-time- or probe-concentration-dependent manner. Therefore, compound 90a can be expected to be a an essential photophore for photolabeling and fluorescent imaging of target molecules without the necessity of preinstalling a large, unstable fluorophore in the probe.

Hotta, Y. et al. developed a novel diazirinyl-(coumarin-4-yl)methyl ester that possesses multiple photochemical properties in a single molecule: crosslinking, fluorogenicity, and cleavage functions regulated through photoinduced electron transfer (PeT) [108]. The cleavage system especially has a large advantage for higher-resolution mass spectroscopic analysis of labeled peptides because a ligand and a purification tag in the PAL probe often complicate analysis (Figure 2).

Therefore, to evaluate the potential of diazirinyl-(coumarin-4-yl)methyl esters for PAL using carbonic anhydrase II (CA-II), PAL probe 97 was synthesized, and benzenesulfonamide was incorporated as a potent inhibitor of CA-II enzyme activity [109] and biotin. Firstly, compound 92 was subjected to cyclization with ethyl-4-chloroacetoacetate to form coumarin 93, followed by installation of Fmoc-Lys(Boc)-OH with ester linkage to give 94. Next, the Fmoc group of 94 was removed using acidic conditions; then, the amino group of 95 was coupled with NHS-biotin to generate 96 with a moderate yield. Finally, the Boc group in the side chain of 96 was deprotected with TFA/CHCl_3_, followed by the incorporation of 4-sulfamoylbenzoic acid with 1-ethyl-3-(3-dimethylaminopropyl)carbodiimide hydrochloride (EDC-HCl) and 1-hydroxybenzotriazole (HOBt) to yield the desired 97 (Figure 13). The inhibitory activity of 97 towards CA-II was calculated with a p-nitrophenyl acetate (4-NPA) esterase assay [110], and the IC_50_ value was 3.0 μM. The PAL probe 97 was furthermore applied for PAL experiments with CA-II. The photocrosslinked CA-II with a 365 nm wavelength was subjected to purification with the avidin–biotin system, followed by photocleavage with a 313 nm wavelength, and the captured protein was observed at approximately 29 kDa, as was detected with coumarin fluorescence. Therefore, diazirinyl-(coumarin-4-yl)methyl esters succeeded in protein identification with controlled photocrosslinking and photocleavage using different photoirradiation wavelengths.

## 9. Diazirinyl-Substituted Indoles

Indole, an electron-rich heteroaromatic compound, exhibits a wide range of pharmacological activities and is a versatile pharmacophore. In particular, indole derivatives occur broadly in nature; for example, indole-3-acetic acid (auxin) is a cell-growth hormone essential for cellular division and expansion in plants [111], and 2-amino-3-(1H-indol-3-yl)propanoic acid (tryptophan) is an essential amino acid used as a building block in biomolecules. Furthermore, natural indole alkaloids are crucial molecules for drug discovery, and their actions include muscle-contraction, migraine-relief [112], cytotoxic, antibacterial, antimicrobial and antineoplastic activity [113]. Moreover, synthetic indole alkaloids exhibit anti-HIV [114], anti-Alzheimer’s-disease [115], anti-inflammatory, and analgesic effects. Indole and its derivatives are essential; therefore, using them with PAL will contribute more to medicinal chemistry and drug discovery. We have succeeded for the first time in synthesis and postfunctional synthesis of diazirine indoles, especially for diazirinyl indole-3-acetic acid, and we have also evaluated their bioactivity experiments [116]. Subsequently, 5- and 6- bromoindoles 98a and 98b were deprotonated with KH, followed by a lithium–bromide exchange with *t*-BuLi, and nucleophilic substitution with trifluoroacetyl donors afforded trifluoroacetyl indole derivatives 99a and 99b with an approximately 80% yield. The trifluoroacetyl group was converted to oximes 100a and 100b with hydroxylamine hydrochloride in pyridine, followed by tosylation with tosyl chloride in triethylamine and acetone at 0 °C. However, tosyl oximes 101a and 101b were too unstable under silica-gel conditions to be hydrolyzed to the ketone moiety. In order to avoid a decrease in yield, these tosyl oximes were directly converted to diaziridines 102a and 102b without purification. The isolated yield for 102a and 102b was maintained at over 85%. Finally, oxidation of the diaziridine moiety with iodine in triethylamine occurred simultaneously with iodination at the 3-position of the indole skeleton (103a and 103b). Moreover, oxidation with activated MnO_2_ was able to proceed without any side reactions to obtain 104a and 104b with a good yield (80–92%) (Figure 14). Soon after, 6-diazirinyl indole was reported, with a similar synthetic scheme, by Wartmann, T. and Lindel, T. [117].

Furthermore, we developed the postfunctional derivatization of diazirinyl indole for expansion of the use of its derivatives in PAL. Tryptophan is one of the most biologically significant metabolites synthesized from indole. Once, we tried to synthesize diazirinyl tryptophan from TPD according to reported information [118,119,120]; however, the diazirinyl tryptophan could not be obtained due to decomposition of the diazirinyl moiety under the harsh conditions required for its construction. Moreover, tryptophan has been prepared through condensation of indole and serine in acetic acid and acetic anhydride under reflux conditions [121]. However, this reaction was conducted under reflux conditions in the presence of all reagents, which the diazirinyl moiety cannot tolerate. Thus, we modified the reaction conditions; initially, active varieties were generated from serine, acetic anhydride, and acetic acid in reflux conditions without indole, followed by the addition of diazirinyl indoles 104a and 104b to the mixture at a low temperature to yield *N*-acetyl tryptophan derivatives 105a and 105b without decomposition of the diazirinyl moiety. Subsequently, racemates of diazirinyl *N*-acetyltryptophans 105a and 105b were subjected to enzymatic resolution with L-acylase to afford optically pure diazirinyl L-tryptophans 106a and 106b without decomposition of the diazirinyl moiety (Figure 15A). In addition to tryptophan, diazirinyl carbinols 108a and 108b have shown anticarcinogenic, antioxidant, and antiatherogenic effects, and diazirinyl gramines 109a and 109b, which play a defensive role in plants, were able to be synthesized (Figure 15B,C).

Indole-3-acetic acid (IAA), well-known as the plant hormone auxin, is essential throughout cellular division and expansion in plants. However, deeper understanding of the biological mechanism of IAA is required. Thus, diazirinyl IAA is a promising chemical tool for understanding those veiled mechanisms in PAL. Diazirinyl indoles 104a and 104b were reacted with oxalyl chloride, followed by methanolysis, to obtain methyl indole 3-oxoacetates 110a and 110b. Reduction with triethylsilane in trifluoroacetic acid [122] occurred not only on the aromatic α-keto moiety of 110 but between the 2- and 3-positions of the indole skeleton, producing 111a and 111b. Next, dehydrogenation at the 2,3-positions of 111a and 111b with MnO_2_ produced only 113b because 111a was unstable under these conditions. Therefore, to construct 5-diazirinyl IAA methyl ester 113a, compound 110a was subjected to reduction with sodium borohydride to obtain α-hydroxy ester 112a, followed by dehydration with P_2_I_4_ [123] to obtain 5-diazirinyl IAA methyl ester 113a with a moderate yield. This method was also able to be applied to the 6-diazirinyl indoles (from 112b to 113b). Finally, hydrolysis of the methyl ester under alkaline conditions produced the desired 5- and 6-diazirinyl IAA derivatives, 114a and 114b (Figure 16). We also performed oat coleoptile segment growth bioassays using 114a and 114b and compared them to IAA [124]. Typical auxin responses were observed in the presence of 114a and of 114b. Especially, photoreactive IAA-substituted diazirin at the 5-position (114a) exhibited a higher effect than did IAA. Therefore, both photoreactive IAA 114a and 114b will contribute to future elucidation of the role of IAA and its target proteins via PAL.

## 10. Conclusions

Heterocyclic aromatics are very important mother skeletons in the fields of materials science and medicinal chemistry. In synthetic chemistry, for heteroaromatic polymers, diazirines have the potential to overcome previous challenges. In particular, application of heteroaromatic diazirines to materials science requires a thorough understanding of their physical properties and characteristics, such as thermal properties and photostability. Although the application of TPD to materials science is known to some extent, there are few reports on heteroaromatic diazirines, and a review regarding the total synthesis and applications of heteroaromatic diazirines has not been published thus far. As shown in this review, heteroaromatic diazirines exhibit different physical properties to those of TPD, which may lead to breakthroughs in novel-polymer-material creation and other challenges. Therefore, it is suggested that the heteroaromatic diazirines presented here could also contribute to development of sustainable new materials. Furthermore, over the past decades, advances in heteroaromatic medicinal chemistry have stimulated progress in the fields of chemical biology and led to significant improvement in quality of life and an increase in the length of human life. However, there are still many unmet medical needs that cannot be addressed with existing therapeutics. In particular, current drug discovery is a search for hit compounds from huge compound libraries and repositioning of existing drugs. Heteroaromatic compounds are often included in these hit compounds, and understanding their mechanisms of action is a crucial issue in drug discovery. Therefore, heteroaromatic diazirines can make a significant contribution to elucidation of such functional mechanisms in medicinal chemistry. This review provides a comprehensive overview of them. Diazirine-substituted heterocyclic aromatics are expected to be one of the most powerful tools to solve these issues, and various studies that will investigate them are expected and will expand greatly in the future. Therefore, this review will greatly contribute to academia as well as the industry.

## Data Availability

Not applicable.

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
