# Peer review of "Heteroaromatic Diazirines Are Essential Building Blocks for Material and Medicinal Chemistry"

_molecules, 2023, doi:10.3390/molecules28031408_

Round 1

Reviewer 1 Report

The manuscript concern some aspects of diazirines physicochemistry. The organisation of this review is clear and readable for readers. The quality of sketchs is excellent. The paper is written very well. I dont have any major remarks. In my opinion, the paper "Heteroaromatic Diazirines are Essential Building Blocks for 2 Material and Medicinal Chemistry", after some minor revisions:
+ keyword "material science" is rather trivial.
+ please check to format of the references block.
+ conclusion can be expanded with more important details.

Author Response

Response To Reviewer#1

The manuscript concern some aspects of diazirines physicochemistry. The organisation of this review is clear and readable for readers. The quality of sketchs is excellent. The paper is written very well. I dont have any major remarks. In my opinion, the paper "Heteroaromatic Diazirines are Essential Building Blocks for 2 Material and Medicinal Chemistry", after some minor revisions:

+ keyword "material science" is rather trivial.

+ please check to format of the references block.

+ conclusion can be expanded with more important details.

>Our response:

Thanks for the suggestion. According to the Reviewer’s suggestion, we removed the keyword "material science" Also, we have added the new sentence “Heterocyclic aromatics are very important mother skeletons in the fields of materials science and medicinal chemistry. In synthetic chemistry, for hetroaromatic polymers, the diazirines have the potential to overcome previous challenges. In particular, the appli-cation of heteroaromatic diazirines to materials science requires a thorough understanding of their physical properties and characteristics, such as their thermal properties and photostability. Although the application of TPD to materials science is known to some extent, there are few reports on heteroaromatic diazirines, and a review regarding the total synthesis and applications of heteroaromatic diazirines has not been published thus far. As shown in this review, heteroaromatic diazirines exhibit different physical prop-erties to those of TPD, which may lead to breakthroughs in the creation of novel polymer materials and other challenges. Therefore, it is suggested that the heteroaromatic dia-zirines presented here could also contribute to the development of sustainable new materials. Furthermore, over the past decades, advances in heteroaromatic medicinal chemistry have stimulated progress in fields of chemical biology and has led to a sig-nificant improvement in quality of life and an increase in the length of human life. However, there are still many unmet medical needs that cannot be addressed by existing therapeutics. In particular, current drug discovery is the search for hit compounds from huge compound libraries and repositioning existing drugs. Heteroaromatic compounds are often included in these hit compounds and understanding their mechanisms of action is a crucial issue in drug discovery. Therefore, heteroaromatic diazirines can make a significant contribution to the elucidation of such functional mechanisms in medicinal chemistry. This review provides a comprehensive overview of them. The diazir-ine-substituted heterocyclic aromatics are expected to be one of the most powerful tools to solve these issues, and various studies investigating them are expected and will expand greatly in the future. Therefore, this review will greatly contribute to academia as well as industry.” in Conclusions (page 19, line 488-page 20, line 512).

Reviewer 2 Report

Summary of the key contribution of the paper:

The Review of Heteroaromatic Diazirines is Essential Building Blocks for Material and Medicinal Chemistry is explained the heterocyclic aromatics are very important mother skeletons in the fields of materials science and medicinal chemistry also their elucidation will greatly contribute to the development of new functional materials and drugs. The scope of the review is very interesting, the review can be accepted.

The topic is relevant to material science and medicinal chemistry, addressing the scope, limitations and challenges associated with the synthesis of  (3-trifluoromethyl)diazirine-substituted heteroaromatics and its applications to probe or elucidate several biological processes. However, it does not elaborate more on the applications of these heteroaromatics in material science. It is hard to compare this review with other published work. In my opinion this article focuses more on the application and development of  (3-trifluoromethyl)diazirine-substituted heteroaromatics.

This review does not compare and contrast various methodologies or discuss any specific one. It explains almost all the major methods for the introduction or synthesis of (3-trifluoromethyl)diazirine-substituted heteroaromatics in the fields of material science and medicinal chemistry. However, the authors gave more importance to the synthesis and applications of these aromatics in the field of medicinal chemistry as compared to material science, especially related to polymer and supramolecular chemistry.

The conclusions are consistent with the evidence and arguments. These conclusions answered in detail the main questions raised in this review in the synthesis and applications of various diazirine-substituted heteroaromatics especially the (3-trifluoromethyl)diazirine-substituted heteroaromatics. The references are well referenced and clear.

Highlights:

·        This review clearly articulates that the diazirine-substituted heterocyclic aromatics are expected to be one of the powerful tools to solve these issues, and various research using them is expected to expand greatly in the future.

·         The figures and tables are well referenced and clear.

·        The resulting review will greatly contribute to academia as well as industry.

·        This study shown different the combination of heteroaromatics and (3-trifluoromethyl) diazirine is a promising to create better materials and elucidate the unknown mechanisms of action of bioactive heteroaromatic compounds

·        In this review, comprehensive synthesis of (3-trifluoromethyl) diazirine-substituted heteroaromatics is described.

·        Lowlights:

·        There are no Lowlights in this paper.

Author Response

Response To Reviewer#2

The Review of Heteroaromatic Diazirines is Essential Building Blocks for Material and Medicinal Chemistry is explained the heterocyclic aromatics are very important mother skeletons in the fields of materials science and medicinal chemistry also their elucidation will greatly contribute to the development of new functional materials and drugs. The scope of the review is very interesting, the review can be accepted. The topic is relevant to material science and medicinal chemistry, addressing the scope, limitations and challenges associated with the synthesis of (3-trifluoromethyl)diazirine-substituted heteroaromatics and its applications to probe or elucidate several biological processes. However, it does not elaborate more on the applications of these heteroaromatics in material science. It is hard to compare this review with other published work. In my opinion this article focuses more on the application and development of (3-trifluoromethyl)diazirine-substituted

heteroaromatics. This review does not compare and contrast various methodologies or discuss any specific one. It explains almost all the major methods for the introduction or synthesis of (3-trifluoromethyl)diazirine-substituted heteroaromatics in the fields of material science and

medicinal chemistry. However, the authors gave more importance to the synthesis and applications of these aromatics in the field of medicinal chemistry as compared to material science, especially related to polymer and supramolecular chemistry.

The conclusions are consistent with the evidence and arguments. These conclusions answered in detail the main questions raised in this review in the synthesis and applications of various diazirine-substituted heteroaromatics especially the (3-trifluoromethyl)diazirinesubstituted heteroaromatics. The references are well referenced and clear.

Highlights:

  • This review clearly articulates that the diazirine-substituted heterocyclic aromatics are expected to be one of the powerful tools to solve these issues, and various research using them is expected to expand greatly in the future.
  • The figures and tables are well referenced and clear.
  • The resulting review will greatly contribute to academia as well as industry.
  • This study shown different the combination of heteroaromatics and (3-trifluoromethyl) diazirine is a promising to create better materials and elucidate the unknown mechanisms of action of bioactive heteroaromatic compounds
  • In this review, comprehensive synthesis of (3-trifluoromethyl) diazirine-substituted heteroaromatics is described.

  • Lowlights:
  • There are no Lowlights in this paper.

>Our response:

Thanks for the suggestion. According to the Reviewer’s suggestion, we have added the new sentence “Heterocyclic aromatics are very important mother skeletons in the fields of materials science and medicinal chemistry. In synthetic chemistry, for hetroaromatic polymers, the diazirines have the potential to overcome previous challenges. In particular, the appli-cation of heteroaromatic diazirines to materials science requires a thorough understanding of their physical properties and characteristics, such as their thermal properties and photostability. Although the application of TPD to materials science is known to some extent, there are few reports on heteroaromatic diazirines, and a review regarding the total synthesis and applications of heteroaromatic diazirines has not been published thus far. As shown in this review, heteroaromatic diazirines exhibit different physical prop-erties to those of TPD, which may lead to breakthroughs in the creation of novel polymer materials and other challenges. Therefore, it is suggested that the heteroaromatic dia-zirines presented here could also contribute to the development of sustainable new materials. Furthermore, over the past decades, advances in heteroaromatic medicinal chemistry have stimulated progress in fields of chemical biology and has led to a sig-nificant improvement in quality of life and an increase in the length of human life. However, there are still many unmet medical needs that cannot be addressed by existing therapeutics. In particular, current drug discovery is the search for hit compounds from huge compound libraries and repositioning existing drugs. Heteroaromatic compounds are often included in these hit compounds and understanding their mechanisms of action is a crucial issue in drug discovery. Therefore, heteroaromatic diazirines can make a significant contribution to the elucidation of such functional mechanisms in medicinal chemistry. This review provides a comprehensive overview of them. The diazir-ine-substituted heterocyclic aromatics are expected to be one of the most powerful tools to solve these issues, and various studies investigating them are expected and will expand greatly in the future. Therefore, this review will greatly contribute to academia as well as industry.” in Conclusions (page 19, line 488-page 20, line 512).

Reviewer 3 Report

Heteroaromatic Diazirines are Essential Building Blocks for 2 Material and Medicinal Chemistry

The review covers various synthetic methods for 3-trifluoromethyl diazirines bearing heteroaromatic, which are promising to create better materials and mechanism elucidation. The synthetic methods are well organized, including complex molecules as shown in Scheme 13. It is obvious that the review is very useful for the broad reader not only organic chemist but also biologist. Therefore I recommend that this review is suitable for the publication in Molecules without any change.

Author Response

Response To Reviewer#3

“Heteroaromatic Diazirines are Essential Building Blocks for 2 Material and Medicinal Chemistry” The review covers various synthetic methods for 3-trifluoromethyl diazirines bearing heteroaromatic, which are promising to create better materials and mechanism elucidation. The synthetic methods are well organized, including complex molecules as shown in Scheme 13. It is obvious that the review is very useful for the broad reader not only organic chemist but also biologist. Therefore I recommend that this review is suitable for the publication in Molecules without any change.

>Our response:

Thank you very much for your very positive comments to our review paper.